# Multi-Time Point Transcriptome Analysis and Functional Validation Revealed *Bol4CL41* Negatively Regulates Black Rot Resistance in Cabbage

**DOI:** 10.3390/ijms26136179

**Published:** 2025-06-26

**Authors:** Hongxue Ma, Siping Deng, Congcong Kong, Yulun Zhang, Tong Zhao, Jialei Ji, Yong Wang, Yangyong Zhang, Mu Zhuang, Limei Yang, Marina Lebedeva, Vasiliy Taranov, Anna M. Artemyeva, Zhiyuan Fang, Jingquan Yu, Zhangjian Hu, Honghao Lv

**Affiliations:** 1Department of Horticulture, College of Agriculture and Biotechnology, Zhejiang University, Hangzhou 310058, China; mahongxue2020@163.com (H.M.); fangzhiyuan@caas.cn (Z.F.); jqyu@zju.edu.cn (J.Y.); 2State Key Laboratory of Vegetable Biobreeding, Institute of Vegetables and Flowers, Chinese Academy of Agricultural Sciences, Beijing 100081, China; dsp2305491314@163.com (S.D.); 13121238399@163.com (C.K.); zhangyulun995@163.com (Y.Z.); 13065207061@163.com (T.Z.); jijialei@caas.cn (J.J.); wangyong03@caas.cn (Y.W.); zhangyangyong@caas.cn (Y.Z.); zhuangmu@caas.cn (M.Z.); yanglimei@caas.cn (L.Y.); 3Department of Plant Pathology, China Agricultural University, Beijing 100083, China; 4All-Russia Research Institute of Agricultural Biotechnology, Russian Academy of Sciences, 119991 Moscow, Russia; arctagrostis@gmail.com (M.L.); v.taranov1@gmail.com (V.T.); 5Federal Research Center N.I. Vavilov All-Russian Institute of Plant Genetic Resources, 190000 St. Petersburg, Russia; akme11@yandex.ru

**Keywords:** multi-time point transcriptome, cabbage, black rot, *4CL* gene family, functional validation

## Abstract

4-coumarate-CoA ligase (4CL) plays a crucial role in the phenylpropanoid metabolic pathway and is a key enzyme involved in plant growth and stress responses. Black rot, caused by *Xanthomonas campestris* pv. *campestris* (*Xcc*) is a major bacterial disease affecting the production of global cruciferous crop-like cabbage (*Brassica oleracea* var. *capitata*). However, the role of *4CL* genes in cabbage resistance to black rot remains unclear. In this study, transcriptome sequencing was conducted using resistant cabbage MY and susceptible cabbage LY at 0, 6, 24, and 48 h post-inoculation. KEGG analysis identified the enrichment of the phenylpropanoid biosynthesis pathway, and significant expression changes of *4CL* genes were determined through the expression heat map. Further genome-wide analysis revealed 43 *Bol4CL* gene family members on the cabbage genome distributed across nine chromosomes. Gene structure and protein motif analysis revealed similarities in motifs within the same evolutionary branch, but variations in gene structure. A combination of *Bol4CL* gene expression profiles and differentially expressed genes (DEGs) from the transcriptome identified *Bol4CL41* as a key gene for further study. Inoculation of overexpressed *Bol4CL41* T_2_ generation stably expressed cabbage seedlings demonstrated significantly larger lesion areas compared to wild type cabbage, indicating that *Bol4CL41* negatively regulates resistance to black rot in cabbage. The analysis of multi-time point transcriptomes in cabbage and the functional study of the *Bol4CL* gene family enhance our understanding of the mechanisms underlying plant disease resistance. This provides compelling evidence and experimental support for elucidating the mechanisms of black rot resistance in cabbage.

## 1. Introduction

Black rot (BR), caused by the bacterial pathogen *Xanthomonas campestris* pv. *campestris* (*Xcc*), is a major disease affecting the production of cruciferous crops [1]. Due to the limited availability of resistance sources and a lack of clear understanding of the molecular mechanisms of disease resistance in cabbage, BR inflicts significant damage. Up to now, BR has spread worldwide, especially causing severe damage to cabbage and other cruciferous vegetables in Asia, Europe, and North America, leading to yield reductions exceeding 70% in severe cases [2,3,4,5]. BR is a vascular disease that mainly affects the leaves of the host plant, from the seedling stage to maturity. The pathogen can invade the plant through hydathodes, stomata, and wounds under suitable conditions of temperature and humidity, resulting in localized tissue necrosis. In severe cases, plant growth is stunted, leaves become brittle and dry, and the entire plant may wilt, die, and appear scorched [1,6,7]. Given the detrimental impact of BR on cabbage and other cruciferous vegetables, it is imperative to investigate the mechanisms of plant disease resistance.

The phenylpropanoid metabolic pathway is essential for plant survival, supplying plants with a large number of secondary metabolite precursors that contribute to growth, development, and resistance to external environmental factors [8]. The 4-coumarate-CoA ligase (4CL) is a key enzyme in this pathway, encoded by a multigene family in higher plants, with the number of gene members varying among different species [9]. Since the cloning of the first *4CL* gene in 1981 [10], numerous *4CL* and *4CL*-like genes have been identified across various plant species. For example, in *Arabidopsis*, four *4CL* genes and nine *4CL*-like genes have been identified [11], five family members have been found in rice [12], and 20 family members in tobacco [9]. *4CL* belongs to the adenosine monophosphate (AMP)-binding protein family [11]. Based on the functions of the proteins encoded by *4CL* genes, *4CL* can be divided into three subgroups. Class I is mainly involved in regulating the biosynthesis of plant lignin compounds, and Class II mainly regulates flavonoid formation [13]. Class III is *4CL*-like, which has different gene functions from the first two subgroups, and lacks the ability to catalyze the formation of CoA esters from substrates [14].

*4CL* members affect plant growth, the biosynthesis of phenylpropanoid derivatives, and responses to environmental stress, effectively regulating and enhancing plant–environment interactions [15]. The *4CL* gene family has been identified as a class of stress resistance genes [16], playing a crucial role when many plants are subjected to abiotic stresses. Ehlting et al. [17] found that overexpressing *4CL* can enhance drought tolerance in *Arabidopsis*. Transcriptional profiling analyses indicated that the expression of *4CL2*, *4CL11*, and *4CL12* changed significantly in one or both desert poplars in response to salt stress compared to that in *P. trichocarpa*. It was hypothesized that the evolution of the *4CL* genes may have promoted the development of both desert poplars [9]. *Me4CL32* gene changes under various abiotic stresses (drought, salt, cold, heat) and hormone stimulation (GA3 and ABA), indicating that *Me4CL32* can respond to both environmental stresses and hormonal signals [18]. The *4CL* gene family also plays an important role under biotic stresses. After *Arabidopsis* was infected with downy mildew, the accumulation of *AtCL1/2* mRNA was induced after 12 and 24 h, respectively [17]. When tomatoes were infected with *Alternaria solani*, the transcriptional levels of *Sl4CL* genes were found to be up-regulated [19]. These data suggest that in plants, *4CL* plays a significant role in both biotic and abiotic stress responses.

Cabbage is one of the most important species in the Brassicaceae family [20], but its yield is severely affected by BR. Therefore, investigating the mechanisms of disease resistance in cabbage is of considerable importance. In this study, we analyzed the transcriptome data of extremely resistant and susceptible cabbage varieties at various stages after inoculation, focusing on the key enzyme 4CL in the phenylpropanoid synthesis pathway. We identified members of the *4CL* gene family in cabbage and determined their phylogenetic relationships, conserved motifs, chromosomal locations, synteny, and expression profiles of the *Bol4CL* genes at different stages following inoculation. Notably, we identified *Bol4CL41* as a key gene exhibiting differential expression between resistant and susceptible materials, and functional validation revealed that *Bol4CL41* negatively regulates cabbage resistance to BR.

## 2. Results

### 2.1. Comparison of the Phenotypes of MY and LY After Inoculation

To compare BR resistance between MY and LY, we first inoculated the two materials at the seedling stage. Leaf tissues were collected pre-inoculation (0 h) and at 6, 24, and 48 h post-inoculation (hpi), followed by multi-time point transcriptome sequencing. A resistance assessment at 14 d post-inoculation revealed minimal lesions on MY (Figure 1a), with no disease symptoms observed on the inner leaves. In contrast, LY exhibited systemic disease, characterized by larger and deeper lesions. Statistical analysis of the lesion area (Figure 1b) demonstrated a significantly larger lesion area in LY compared to MY. This makes them excellent research materials for investigating the mechanisms of BR resistance.

### 2.2. Functional Annotation of DEGs and Expression of Phenylpropanoid Biosynthesis Genes

Differential expression gene analysis was performed on 10 combinations: R0 vs. S0, R6 vs. S6, R24 vs. S24, R48 vs. S48, R0 vs. R6, R0 vs. R24, R0 vs. R48, S0 vs. S6, S0 vs. S24, and S0 vs. S48, (Figure 2a). A total of 26 differentially expressed genes were common to all 10 groups (Appendix A), of which 2 genes (*BolC04g001130.2J* and *BolC06g036850.2J*) contained an AMP-binding enzyme domain and exhibited consistent differential expression trends in 8 groups.

To investigate the influence of intrinsic differences between MY and LY on cabbage resistance to BR, we performed GO functional enrichment analysis on DEGs from R0 vs. S0 (Figure 2b). The results showed significant enrichment of GO functions in Biological Process; Biological Process was mainly enriched to respond to external biotic stimulus and glucosinolate biosynthetic process; Cellular Component was mainly enriched in cell periphery, plasma membrane, anchored component of membrane and protein-DNA complex; Molecular Function was enriched for pigment binding. Enrichment results for other groups can be found in Appendix A. To further explore the biological functions of these DEGs, KEGG enrichment analysis was performed (Figure 2c–g and Appendix A). A total of 20 pathways were significantly enriched in each group, including glucosinolate biosynthesis, fatty acid degradation, phenylpropanoid biosynthesis, and flavonoid biosynthesis. The phenylpropanoid biosynthesis pathway was found to be enriched in five groups: R0 vs. S0, R24 vs. S24, R0 vs. R24, S0 vs. S24, and S0 vs. S48 (Figure 2c–g). Different resistant plants have certain variability, and also according to the electron microscope observation of Ma et al. [21], the pathogens gathered on the plant leaves at 24 hpi, and the pathogens proliferated and invaded the plants at 48 hpi, so very critical time nodes are at 0 h, and 24, 48 hpi. Besides that, two of the above genes that were cross-differentially expressed in 10 combinations were in the phenylpropanoid biosynthesis pathway, so we screened three highly expressed genes involved in the phenylpropanoid biosynthesis pathway from the transcriptome data and plotted the expression heat map (Figure 2h). In the phenylpropanoid biosynthesis pathway, the expression of *4CL* in MY and LY varied greatly, with lower expression in MY and higher expression in LY. It was hypothesized that *4CL* may contribute to the susceptibility of cabbage to disease; therefore, the *4CL* gene family in cabbage will be the focus of subsequent research.

### 2.3. Systematic Identification and Phylogenetic Analysis Reveal Functional Divergence in the Cabbage 4CL Gene Family

A genome-wide analysis identified 43 *Bol4CL* genes in the cabbage JZS V2.0 genome, which were named *Bol4CL1-Bol4CL43* (Appendix A). The length of protein ranged from 352 amino acids (AAs) (Bol4CL43) to 1543 (Bol4CL36) amino acids, and the predicted molecular weights varying from 38.36 kDa (Bol4CL43) to 169.61 kDa (Bol4CL36). The theoretical pI varied from 4.64 (Bol4CL36) to 8.91 (Bol4CL27). The subcellular location was predicted, which provided possible clues for functional research. The results of subcellular localization prediction showed that Bol4CLs were mainly located in the cytoplasm (Bol4CL12, Bol4CL16, Bol4CL18, Bol4CL22, Bol4CL24, Bol4CL25, Bol4CL26, Bol4CL29, Bol4CL33, Bol4CL41, Bol4CL42, and Bol4CL43). This suggests that this gene family may play a potentially significant role in the cytoplasm.

In order to determine the evolutionary relationships between *Bol4CLs* and *4CLs* to other species, we performed multiple sequence alignment and generated a NJ phylogenetic tree for *4CLs* from *Arabidopsis*, rice, broccoli, and Chinese cabbage (Figure 3). A gene ID renaming table can be found in Appendix A. For this study, we generated a version of the phylogenetic reconstruction that incorporated the 4CL and 4CL-like amino acid sequence data from cabbage, *Arabidopsis*, rice, broccoli, and Chinese cabbage. In total, all *4CLs* from 5 species are clustered into three classes, designated as Class 1, Class 2, and Class *4CL*-likes. Class 1 contained three *At4CL* genes (*At4CL1*, *At4CL2*, and *At4CL3*) and four *Os4CL* genes (*Os4CL1*, *OsAt4CL3*, *Os4CL4*, and *Os4CL5*), Class 2 contained *At4CL4*, *Os4CL5*, and *Bol4CL41*, while Class *4CL*-likes included all of the *4CL*-like genes identified from *Arabidopsis* (*At4CL5*, *At4CL6*, *At4CL7*, *At4CL8*, *At4CL9*, *At4CL10*, *At4CL11*, *At4CL12*, and *At4CL13*) and mostly derived from the genes of cabbage, broccoli, and Chinese cabbage. Based on evolutionary relationships, *Os4CL* genes are found only in Classes 1 and 2, while *Bol4CL* genes are present in Classes 2 and *4CL*-likes. *Arabidopsis*, broccoli, and Chinese cabbage are distributed across Classes 1, 2, and *4CL*-likes. There is potential functional similarity among the *4CL* genes within the same group; the phylogenetic tree results help predict the functions of *Bol4CL* genes.

### 2.4. Gene Structure and Conserved Motif of Bol4CLs

To better understand the structural diversity of *Bol4CL* genes, conserved protein motifs were identified using the MEME online tool. A total of 10 putative conserved motifs were identified within the *Bol4CL* genes. The number of motifs identified in the Bol4CLs varied considerably, ranging from 4 to 9. Motifs 5 and 6 were present in all Bol4CLs, while motif 1 was found in 42 *Bol4CLs* and motif 3 in 39. Conserved domains within the same phylogenetic branch exhibited structural similarities (Figure 4a). To better understand the evolution of the *Bol4CL* gene family, we analyzed the gene structure. The coding sequence (CDS)-intron structure is shown in Figure 4b. Among the 43 *Bol4CLs* analyzed, the number of CDSs in *Bol4CLs* is different. Furthermore, differences in the lengths of introns and exons were observed among the members of the *Bol4CL* family.

### 2.5. Chromosomal Location and Synteny Analysis of Bol4CLs

Synteny analysis can reveal the evolutionary history and functional conservation of gene families. Therefore, we investigated the synteny of *Bol4CL* genes to explore their functions and regulatory mechanisms in cabbage and other species. Through this analysis, we identified 15 homologous gene pairs among the 43 members of the *Bol4CL* gene family located on the 9 chromosomes of cabbage, specifically: *Bol4CL20*/*Bol4CL6* and *Bol4CL10*, *Bol4CL20*/*Bol4CL29*, *Bol4CL33*/*Bol4CL29*, *Bol4CL8* and *Bol4CL34*/*Bol4CL6* and *Bol4CL10*, *Bol4CL26*/*Bol4CL15*, *Bol4CL24*/*Bol4CL26*, *Bol4CL22*/*Bol4CL33*, *Bol4CL25* and *Bol4CL41*/*Bol4CL28*, *Bol4CL39*/*Bol4CL40*, *Bol4CL2*/*Bol4CL3*, *Bol4CL16* and *Bol4CL18*/*Bol4CL42*, *Bol4CL12*/*Bol4CL42*, *Bol4CL14*/*Bol4CL27*, *Bol4CL12*/*Bol4CL16*, and *Bol4CL18*, *Bol4CL22*/*Bol4CL29* (Figure 5a). Chromosomal location analysis of the cabbage and rice genomes identified few collinear gene pairs, only five. In contrast, collinearity analysis between cabbage and *Arabidopsis* identified more collinear gene pairs (Figure 5b). The collinearity between the cabbage and *Arabidopsis* genomes was significantly higher than that between the cabbage and rice genomes, indicating a close homologous relationship between cabbage and *Arabidopsis*, with one homologous gene exhibiting one-to-many collinearity. This further reveals the conservation and functional divergence of these *4CL* genes across species.

### 2.6. Gene Expression Profiles Heatmap of Bol4CL at Different Stages After Inoculation

Studying the expression levels of *Bol4CL* genes during different inoculation stages in cabbage helps us to understand their role in plant disease resistance. The FPKM values of 43 *Bol4CL* genes were screened from the transcriptome data, and heat map analysis was used to visualize the expression profiles of these genes during disease resistance in cabbage (Figure 6). The results showed that a total of 35 *Bol4CL* genes were expressed at all time points after inoculation in both materials. The transcription levels of *Bol4CL* genes varied among these samples, but interestingly, the expression levels of all *Bol4CL* genes were significantly lower at the R24 stage than at other stages. It was speculated that *Bol4CL* genes negatively regulated the resistance to BR in cabbage, and that 24 h after inoculation was a critical period for the colonization and propagation of the bacteria, so that the expression of *Bol4CL* genes in the resistant material was greatly reduced. Genes such as *Bol4CL13*, *Bol4CL14*, and *Bol4CL35* showed sustained expression and relatively high expression levels at different stages in both materials. *Bol4CL41* gene expression was higher in susceptible material and lower in resistant material, exhibiting a similar expression pattern to *Bol4CL2*, which is in the same branch. In conclusion, these genes may play a role in negatively regulating BR resistance in cabbage, but further functional validation is needed to confirm this hypothesis.

### 2.7. Bol4CL41 Negatively Regulates BR Resistance in Cabbage

To elucidate the biological functions of Bol4CL41 proteins in cabbage, we generated transgenic plants overexpressing *Bol4CL41*. Through continuous self-fertilization, homozygous T_2_ generation plants were ultimately obtained and confirmed the resulting altered gene expression by qRT-PCR (Figure 7a). Two independent transgenic lines exhibiting high levels of Bol4CL41 were selected for further inoculation tests. The seedling phenotype showed that the transgenic lines showed a significantly increased susceptibility to BR compared to the wild type (GL) after inoculation (Figure 7b,c). These results demonstrate that *Bol4CL41* negatively regulates BR resistance in cabbage.

## 3. Discussion

Transcriptome analysis has been employed for years to elucidate plant–microbe interactions [22]. Transcriptome sequencing and differential gene expression analysis at various time points following plant infection are valuable for illustrating the dynamic gene responses within the plant, contributing significantly to understanding disease resistance or susceptibility mechanisms. Research into the gene regulation of responses to pathogen attack may reveal the roles of relevant defense genes [23]. Sun et al. [24] performed transcriptome analysis on resistant and susceptible cabbage varieties at 0, 12, 24, 48, and 96 h post-inoculation, discovering enhanced glucosinolate biosynthetic and catabolic pathways. To investigate the differential expression of *JAZ* genes in cabbage after *Xcc* infection, Liu et al. [25] analyzed RNA-seq data from resistant and susceptible varieties post-inoculation, identifying both induced and repressed *JAZ* genes. Transcriptome analysis of cruciferous leaves at 3 and 12 d post-inoculation revealed that 78 and 809 genes up-regulated, and 10 and 169 genes down-regulated, at early and late stages, respectively. Genes related to terpenes, flavonoids, alkaloids, and anthocyanins and phytohormones were up-regulated at the early stage of infection [26]. Tortosa et al. [27] demonstrated that *CBP60g* and *SARD1* contribute to *Xcc* resistance, as evidenced by transcriptome analysis of broccoli at 3 and 12 dpi and validation in *Arabidopsis* knockout mutants. Certain metabolic and signaling pathways are consistently activated during plant–pathogen interactions. In this study, we performed transcriptome sequencing on two cabbage varieties at 0, 6, 24, and 48 h post-inoculation. Glucosinolate biosynthesis and flavonoid biosynthesis were identified as enriched KEGG pathway; the results of Sun et al. [24] and Tortosa et al. [26] analyses verified this result. Moreover, previous transcriptome analyses have indicated that the expression of resistance-related genes varies at different time points post-inoculation. Our research confirms this, showing that the expression levels of all *Bol4CL* genes were lower at R24 compared to other time points. This suggests that the timing of defense activation is crucial for plant survival, so it is necessary to detect the gene dynamics before and after plant inoculation and cross-analysis of multiple time points.

4CL is an important rate-limiting enzyme in the phenylpropane metabolism pathway [28]. The activity levels of 4CL enzymes significantly influence the accumulation of substances such as flavonoids and lignin. Furthermore, *4CLs* play a crucial role in plant growth and resistance to external environmental stresses [29]. Therefore, the systematic identification and comprehensive analysis of the *4CL* gene family are of great significance for understanding the functions of these family members. The *4CL* gene family at the genome-wide level has been reported in various species [30,31,32,33]. However, the *4CL* family has not been systematically studied in cabbage, an economically and nutritionally vital crop. In this study, 43 *4CL* genes were identified in cabbage and named *Bol4CL1-43*. Cabbage possesses a greater number of *4CL* members compared to *Arabidopsis* [12] and rice [13], likely due to differences in genome size or the evolutionary diversity of these gene families. Phylogenetic reconstruction is a robust tool for resolving gene family evolution [34]. In plants, *4CL* genes can be categorized into three subgroups: Class I, II, and the *4CL*-like subgroup [10]. Based on the classification of *At4CL* genes, this study also divided *Bol4CL* genes into three subgroups. Class I and II contain fewer members, with *Bol4CLs*, *BolH4CLs*, and *Bra4CLs* predominantly clustered in the Class *4CL*-like subgroup. Variations between different species may be primarily related to their evolutionary level. Further analysis of conserved motifs and gene structures in *Bol4CL* proteins indicated that, while conserved motifs within the same subgroup exhibited similarity, the distribution of introns and exons varied considerably. This variation was exemplified by *Bol4CL12* and *Bol4CL42*, both in the same clade, *Bol4CL42* displaying an exceptionally long intron sequence. There were differences in the structure and motif types of each member gene, which may cause the functional differences among the subgroups. Analyzing gene collinearity across different species has revealed additional insights into the evolutionary relationship between *Bol4CL* genes and *At4CL*, *Os4CL* genes. This analysis enhances our understanding of the phylogenetic tree and broadens our knowledge base.

Increasing studies have shown that *4CLs* play critical roles in the biotic stress responses of plants [35,36]. Analysis of gene expression patterns in plants under biotic stress can provide valuable insights into gene function. Differentially expressed genes may be candidate genes involved in the plant’s stress response. We constructed a clustered heatmap to analyze the expression profiles of 43 *Bol4CL* genes in MY and LY at different time points post-inoculation with *Xcc*. The results revealed that *Bol4CL41* and *Bol4CL2* exhibited higher expression levels in the susceptible material and lower levels in the resistant material, suggesting that *Bol4CL2* and *Bol4CL41* may play a role in negatively regulating BR resistance in cabbage. Furthermore, *Bol4CL41* was identified as a differentially expressed gene across all 10 groups in the transcriptome analysis, making it a key gene for further investigation.

The transcriptional activation of *4CLs* has been demonstrated in cultured cells of various plants, including soybean, parsley, and *Arabidopsis*, following treatment [35,36,37]. In addition, this activation has been observed in soybean and parsley inoculated with *Phytophthora sojae* and in potato infected with *Phytophthora infestans* [37,38,39]. In *Arabidopsis*, a transient accumulation of *4CL* mRNA was detected after infiltrating leaves with an avirulent strain of the bacterial pathogen, *Pseudomonas syringae* pv. *maculicola* [40]. Our transcriptome analysis of the *Bol4CL* gene family at different time points after inoculation with MY and LY (Appendix A) revealed that the expression of most genes in the *Bol4CL* gene family was up-regulated in LY compared to MY, with only a small number of genes being down-regulated. Specifically, the expression levels of the *Bol4CL2*, *Bol4CL18*, *Bol4CL41*, and *Bol4CL42* genes increased at every time point (Figure 8). To investigate whether *Bol4CL41* plays a role in cabbage resistance to BR, we first overexpressed Bol4CL41 and transformed it into cabbage. Following continuous self-pollination and selection, we obtained T_2_ generation overexpressing cabbage seedlings. After inoculation, we found that two lines of OE-*Bol4CL41* cabbage exhibited larger lesion areas compared to wild type cabbage. These results indicate that the *Bol4CL41* gene negatively regulates cabbage resistance to BR.

## 4. Materials and Methods

### 4.1. Plant Materials, Growth Conditions, and Inoculation Method

The cabbage inbred lines were obtained from the Institute of Vegetables and Flowers, Chinese Academy of Agricultural Sciences (IVF-CAAS), Beijing, China. Both MY and LY were bred through resistance identification and systematic selection. MY is a highly resistant line, while LY is highly susceptible. Seedlings were grown in a greenhouse at 24–28 °C with 60–70% relative humidity for one month, watered 3–4 times per week, and inoculated when 4–5 true leaves had grown.

The inoculation strain used was *Xcc* race 1-typed strain WHI3811 [41], provided by Dr. Joann G. Vicente, University of Warwick, UK. *Xcc* was cultured in nutrient broth (NB) medium containing 5 g of peptone, 3 g of beef extract, and 5 g of NaCl per liter of distilled water, with a pH of 6.5–6.8 [42], with shaking at 28 °C and 200 rpm/min for approximately 16 h until it reached the logarithmic phase, after which it was adjusted to a concentration of 1 × 10^8^ cfu/mL with an appropriate amount of sterile water (OD600 = 0.2).

With slight modifications based on the spray inoculation method of Kong et al. (2021) [43], seedlings were sprayed with a fine mist of water and covered with plastic film for 14 h, maintaining an indoor temperature of 15–18 °C and indoor relative humidity above 90%, with relative humidity inside the film maintained above 95%. After the leaf margins were spatulated with water using a medical throat sprayer, the bacterial liquid was evenly sprayed on the cabbage leaves until it covered the leaves in the form of fine water droplets and covered the plastic film. After 24 h, the plastic film was removed while a room temperature of 26–28 °C was maintained.

### 4.2. RNA-Seq and Transcriptome Analysis of Cabbage in Response to BR

MY and LY leaf samples were taken before inoculation (0 h) and 6, 24, and 48 h after inoculation, and quickly placed in liquid nitrogen and stored at −80 °C until RNA extraction. RNA extraction and sequencing were performed by Personalbio Biotechnology Co., Ltd. (Shanghai, China).

Following sequencing, the samples yield image files that are converted using the sequencing platform’s software of Personalbio Biotechnology Co., Ltd. (Shanghai, China) to generate FASTQ raw data, which is the initial output. The sequencing data contains some joints and low-quality reads, which can significantly interfere with subsequent analysis, so it is necessary to further filter the sequencing data. The criteria for data filtering include the following: removing the junction at the 3′ end by Cutadapt 1.18, removing sequences with at least a 10 bp overlap with the known junction (AGATCGGAAG), allowing for 20% base mismatches. The reads with an average quality score lower than Q20 are removed.

Filtered reads were compared to the cabbage reference genome of Brassicaceae Database (BRAD, http://www.brassicadb.cn/#/, accessed on 27 February 2025) using HISAT2 (http://ccb.jhu.edu/software/hisat2/index.shtml, accessed on 27 February 2025) software. The read count value on each gene was counted using HTSeq2.0 [44] as the raw expression of the gene. Reads counts were positively correlated with the true expression levels of the genes, as well as with gene length and sequencing depth. To ensure comparability of gene expression levels across different genes and samples, Fragments Per Kilobase of transcript per Million reads mapped (FPKM) was used to normalize the expression. Differential expression analysis and gene selection are subsequently performed using DESeq2 (https://bioconductor.org/packages/release/bioc/html/DESeq2.html, accessed on 27 February 2025) [45], with differentially expressed genes identified based on the criteria of |log2FoldChange| > 1 and a significance threshold of *p* value < 0.05.

GO (http://geneontology.org/, accessed on 27 February 2025) and KEGG (http://www.kegg.jp/, accessed on 27 February 2025) annotations were used for classification. GO and KEGG enrichment analyses were performed using topGO (https://bioconductor.org/packages/release/bioc/html/topGO.html, accessed on 27 February 2025) [39] and clusterProfiler (https://bioconductor.org/packages/release/bioc/html/clusterProfiler.html, accessed on 27 February 2025) [46], respectively. The *p* value was calculated using the hypergeometric distribution method (the significance threshold for enrichment was *p* value < 0.05) and corrected by FDR to obtain Q value. Generally, functions with Q value < 0.05 were regarded as significant enrichment. To identify GO term and KEGG pathway compared with the whole genomic background, the major biological functions exercised by the differential genes were identified. The KEGG enrichment maps were selected from R0 vs. S0 time points, and −log10(*p* value) > 1.3 was considered as significant enrichment; the 23 GO terms with the smallest Q value in R0 vs. S0 time points were selected for the enrichment maps. Both the GO enrichment plot and the multi-time point expression heatmap of the phenylpropanoid biosynthesis pathway were generated using ChiPlot (https://www.chiplot.online/, accessed on 27 February 2025), with −log10(Q value) > 1.3 considered significantly enriched.

### 4.3. Genome-Wide Identification of 4CL in Different Species

The sequences of the rice *Os4CL* family genes were obtained from the Rice Annotation Project Database (RAP-DB, https://rapdb.dna.affrc.go.jp/, accessed on 27 February 2025), and *Arabidopsis At4CL* family gene sequences were sourced from The *Arabidopsis* Information Resource (TAIR, https://www.arabidopsis.org/, accessed on 27 February 2025). *Bol4CL* (cabbage), *BolH4CL* (broccoli), and *Bra4CL* (Chinese cabbage) family gene sequences were sourced from BRAD (http://www.brassicadb.cn/#/, accessed on 27 February 2025). Initially, InterPro (https://www.ebi.ac.uk/interpro/search/, accessed on 27 February 2025) was used to identify the PFAM ID for the *4CL* gene family as PF00501. Subsequently, the Hidden Markov Model (HMM) (https://www.ebi.ac.uk/interpro/download/Pfam/, accessed on 27 February 2025) was employed, along with TBtools-II (https://github.com/CJ-Chen/TBtools-II, accessed on 27 February 2025) [30] to perform the initial screening of family members. Sequences were extracted using the NCBI CD-Search Tool (https://www.ncbi.nlm.nih.gov/Structure/bwrpsb/bwrpsb.cgi, accessed on 27 February 2025), and genes with an E-value ≤ 1 × 10^−5^ and a bitscore ≥ 200 were selected for secondary screening, ultimately defining the family members.

### 4.4. Construction of Phylogenetic Trees and Analysis of Protein Characteristics

ClustalW alignment was performed in MEGA7 (https://www.megasoftware.net/, accessed on 27 February 2025), and a phylogenetic tree was constructed using the Neighbor-Joining (NJ) method, with 1000 bootstrap replications used to assess tree topology and reliability. Evolview (https://evolgenius.info/helpsite/qst1.html, accessed on 27 February 2025) was used to visualize the phylogenetic tree.

Using the ExPASy ProtParam online tool (https://web.expasy.org/protparam/, accessed on 27 February 2025), the relative molecular weight (MW), theoretical isoelectric point (pI), instability index, aliphatic index, and grand average of hydropathicity (GRAVY) protein characteristics of Bol4CL were predicted. Using the WoLF PSORT online website (https://psort.hgc.jp/, accessed on 27 February 2025), a subcellular localization prediction for the Bol4CL protein was performed. The specific chromosomal location of the gene from the BRAD was obtained (http://www.brassicadb.cn/#/, accessed on 27 February 2025).

### 4.5. Gene Structural Analysis of Bol4CL Genes

To effectively analyze gene structures and motifs, the full-length protein sequences of Bol4CL were uploaded to the online website MEME Suite 5.5.7 (https://meme-suite.org., accessed on 27 February 2025) to search for conserved motifs by finding motif numbers set to up to 10 in sequences. Gene structure annotations in GFF3 format for cabbage were downloaded from the BRAD (http://www.brassicadb.cn/#/, accessed on 27 February 2025). Next, the TBtools-II software [30] was used for the visualization of motifs and gene structures.

### 4.6. Chromosome Location and Synteny Analysis

To investigate the genomic synteny among cabbage, rice, and *Arabidopsis*, synteny analyses were performed using the One Step McScanX-Super Fast module in TBtools software [47]. The genome sequence files and gene structure annotation files for the three species were imported into the software, with E value set to 1 × 10^−5^ and the blast hit count set to 10. Following completion of the synteny analysis, the Multiple Synteny Plot module in TBtools [48] was used to further visualize the results. Visualization of the cabbage genome Circos plot was performed using the Advanced Circos module.

### 4.7. Construction Gene Expression Profiles Heatmap

To study the expression profiles of *Bol4CL* genes across different varieties and various post-inoculation time points, we excavated the transcriptomic data. The transcript abundances were calculated and quantified for each sample using FPKM. Finally, the average FPKM values were log-transformed and used to generate a heatmap using ChiPlot (https://www.chiplot.online/, accessed on 27 February 2025).

### 4.8. Vector Construction, Transgenic Plant Generation and Inoculation Treatments

The overexpression vector pBWA(V)BS-*Bol4CL41* was constructed by inserting the full coding sequence (CDS) of the *Bol4CL41* gene into the transformed vector pBWA(V)BS. For cabbage transformation, the resulting recombinant vector was introduced into cabbage (GL) through the agrobacterium (GV3101)-mediated transformation. Homozygous T_2_ transgenic lines were selected according to their resistance to Basta, and positive lines were identified by qRT-PCR using cDNA. Afterwards, resistance identification was conducted on the T_2_ transgenic positive lines.

### 4.9. Disease Mechanism Diagram Drawing, Resistance Evaluation and Data Analysis

The disease mechanism diagram was created using Figdraw (https://www.figdraw.com/static/index.html#/, accessed on 28 February 2025). The lesion area was calculated using ImageJ V1.8.0.112 software (National Institutes of Health, Bethesda, MD, USA). The data were first calculated in Excel (Beijing Jinshan Office Software Co., Ltd., Beijing, China). Subsequently, the data were statistically analyzed using GraphPad Prism 9.0 software (GraphPad Software Inc., La Jolla, CA, USA). Differences between the two materials were assessed using a *t*-test, with a significance threshold set at *p* < 0.05 and *p* < 0.01 for highly significant differences.

## 5. Conclusions

Following our analysis of multi-time point transcriptomic data, we identified the *4CL* gene family as a key focus. We identified 43 *Bol4CL* genes within the cabbage genome and subsequently analyzed their evolutionary relationships, conserved motifs, gene structures, chromosomal location, and synteny. These analyses provide valuable insights into the functions of this gene family. Gene expression profiles showed that *Bol4CL2* and *Bol4CL41* were differentially expressed between MY and LY, with *Bol4CL41* being a differentially expressed gene among the 10 groups analyzed in the transcriptome analysis. Therefore, a stable genetic transformation line of OE-*Bol4CL41* cabbage was constructed. Inoculation of T_2_ generation seedlings demonstrated that OE-*Bol4CL41* plants were more susceptible to *Xcc* compared to wild type cabbages. These findings indicate that *Bol4CL41* negatively regulates resistance to BR in cabbage, providing compelling evidence and experimental support for elucidating the mechanisms underlying BR resistance.

## Figures and Tables

**Figure 1 ijms-26-06179-f001:**
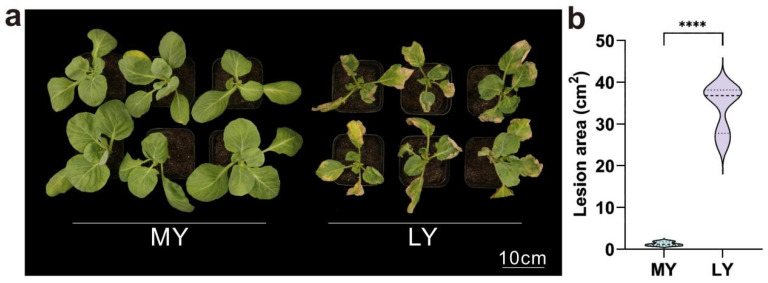
The phenotypes of MY and LY varied greatly after inoculation. (**a**) Phenotypic differences between MY and LY post-inoculation. (**b**) Statistical analysis of the lesion area. The data are shown as the means ± SDs, *n* = 3. Asterisks represent significant differences between two groups according to the Unpaired *t* test (**** *p* < 0.0001).

**Figure 2 ijms-26-06179-f002:**
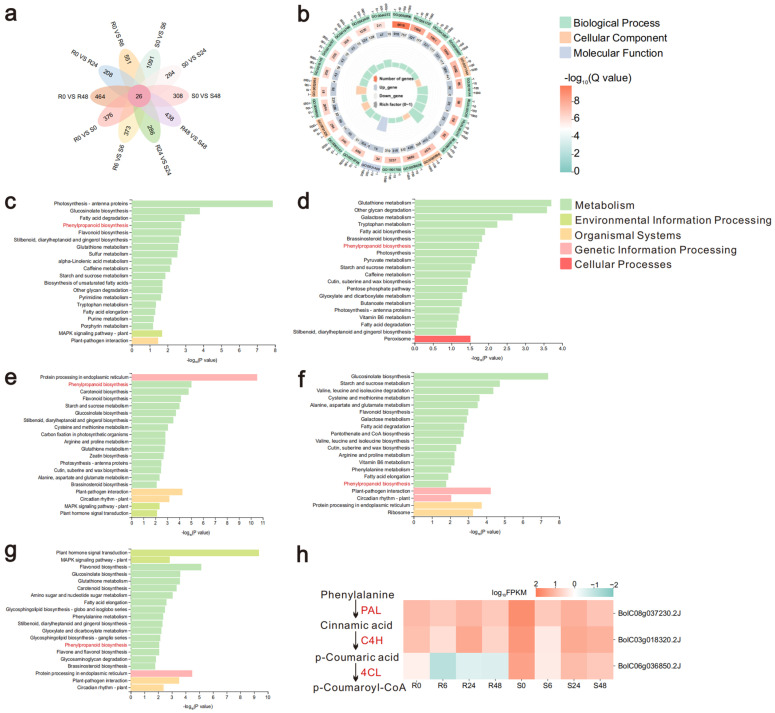
The phenylpropanoid biosynthesis pathway is differentially expressed in cabbage after inoculation. (**a**) Flower plot of 10 sets of differentially expressed genes. (**b**) shows the GO enrichment results for R0 vs. S0. (**c**–**g**) represent KEGG enrichment results for R0 vs. S0, R24 vs. S24, R0 vs. R24, S0 vs. S24, and S0 vs. S48, respectively. (**h**) Expression profiles of key genes in the phenylpropanoid biosynthesis pathway across multiple time points.

**Figure 3 ijms-26-06179-f003:**
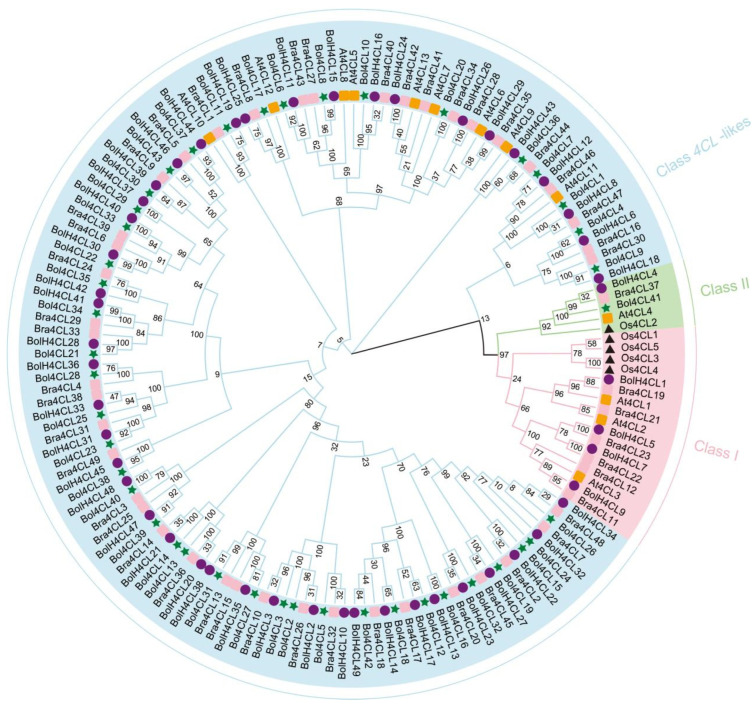
Phylogenetic analysis representing the relationship between 4CL proteins in cabbage, *Arabidopsis*, rice, broccoli and Chinese cabbage. The 4CL subgroup is marked with different colors.

**Figure 4 ijms-26-06179-f004:**
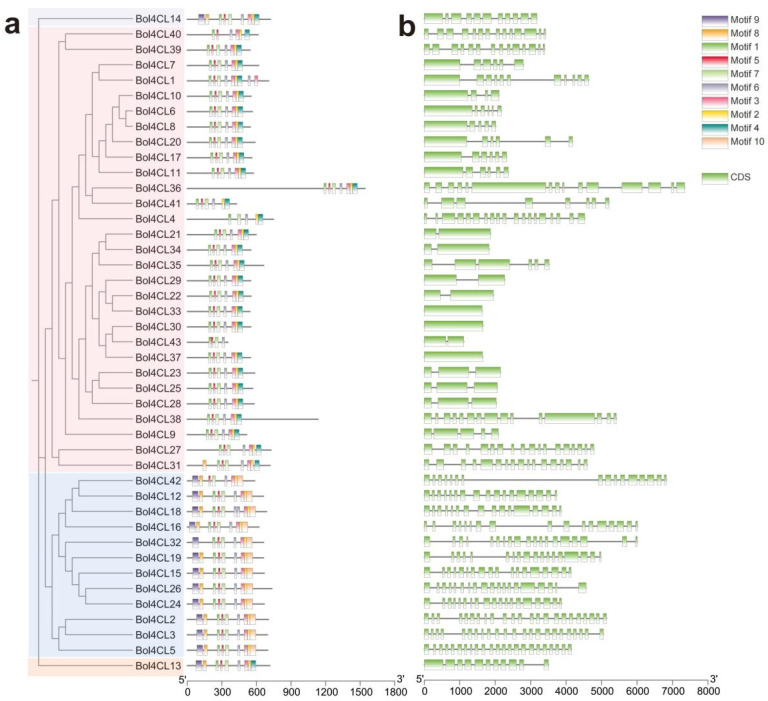
Phylogenetically alignment conserved motifs and gene structures analysis of *Bol4CL* genes in cabbage. (**a**) Different colored boxes indicate the 10 conversed protein motifs of *Bol4CL* genes. (**b**) Gene structure analysis of 43 *Bol4CL* genes.

**Figure 5 ijms-26-06179-f005:**
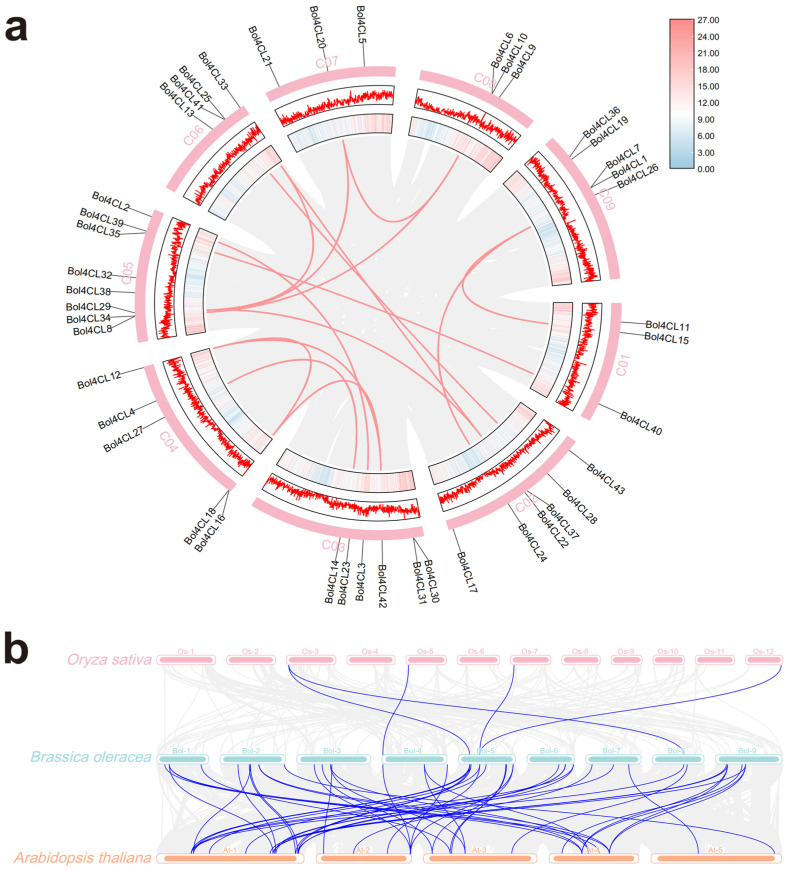
Synteny analysis of *4CL* gene family in different plants. (**a**) Synteny Circos plot within the cabbage genome. The inner circle displays gene density, with the pink color representing higher gene density. The second circle is a linear gene density plot, with the outermost circle representing the chromosome. The red connecting lines indicate syntenic gene pairs. (**b**) Synteny plot between cabbage and rice, cabbage and *Arabidopsis*. The blue connecting lines indicate syntenic gene pairs between the two species.

**Figure 6 ijms-26-06179-f006:**
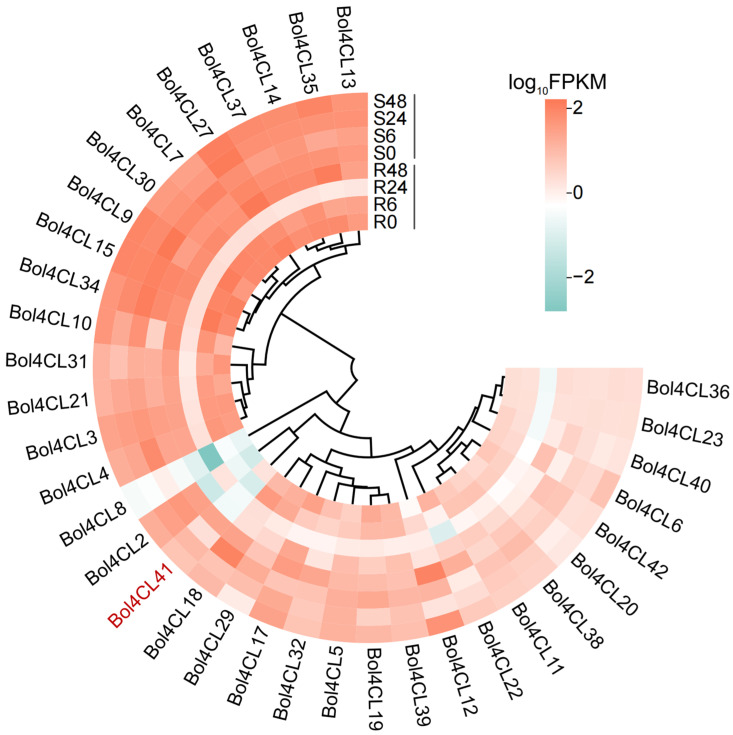
Expression patterns of *Bol4CL* genes differentially expressed in cabbage of different multiple time points post-inoculation.

**Figure 7 ijms-26-06179-f007:**
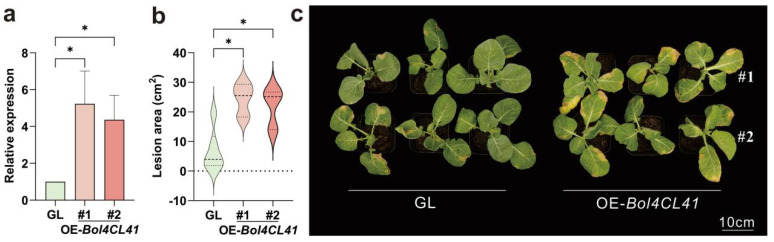
*Bol4CL41* negatively regulates BR resistance in cabbage. (**a**) Expression of GL, OE-*Bol4CL41*#1, and OE-*Bol4CL41*#2. Relative expression of OE-*Bol4CL41*#1 and OE-*Bol4CL41*#2 was calculated relative to GL expression, which was designated as 1. The data are shown as the means ± SDs, *n* = 3. Asterisks represent significant differences across three groups according to Ordinary one-way ANOVA (* *p* < 0.05). (**b**) Statistical analysis of the lesion area. The data are shown as the means ± SDs, *n* = 3. Asterisks represent significant differences across three groups according to Ordinary one-way ANOVA (* *p* < 0.05). (**c**) Phenotypic differences between GL, OE-*Bol4CL41*#1 and OE-*Bol4CL41*#2.

**Figure 8 ijms-26-06179-f008:**
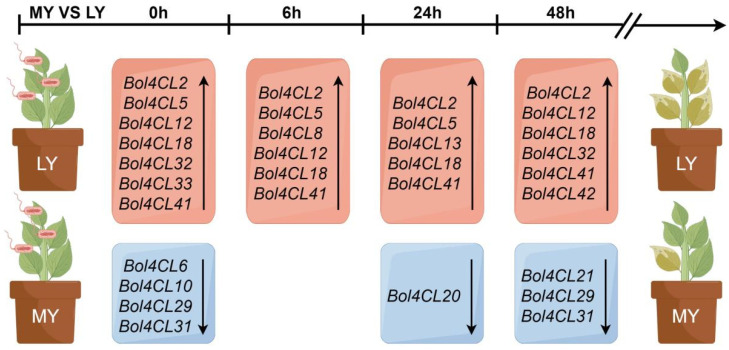
Differential up- and down-regulation of the *Bol4CL* gene family in R0 vs. S0, R6 vs. S6, R24 vs. S24, and R48 vs. S48 contributes to differences in resistance levels to MY and LY.

## Data Availability

All data supporting the findings of this study are available in the paper and its Appendix A published online.

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
