# Peer review of "Multi-Time Point Transcriptome Analysis and Functional Validation Revealed Bol4CL41 Negatively Regulates Black Rot Resistance in Cabbage"

_ijms, 2025, doi:10.3390/ijms26136179_

Round 1

Reviewer 1 Report

Comments and Suggestions for Authors

4-coumarate-CoA ligase (4CL) plays a crucial role in the phenylpropanoid metabolic pathway and is a key enzyme involved in plant growth and stress responses. However, the role of 4CLs in cabbage resistance to black rot remains unclear. This manuscript revealed the key roles of phenylpropanoid biosynthesis pathway mediated by 4CLs in black rot resistance in cabbage via a multi-time point transcriptomes assay. Furthermore, this study revealed that Bol4CL41, a differentially expressed gene from the transcriptomes assay, functions as a negative regulator in black rot resistance in cabbage. I believe this manuscript is interesting. I, therefore, would recommend a minor revision with the following points:

Lines 134-136:Why the phenylpropanoid biosynthesis pathway was found to be only enriched in the five groups (R0 VS S0, R24 VS S24, R0 VS R24, S0 VS S24 and S0 VS S48), why not in another five groups? Please explain and discuss it.

Has the function of Bol4CL2 in BR resistance been verified, given that similar to Bol4CL41 this gene also shows significant expression differences between resistant and susceptible materials?

Why is the function of Bol4CL41 in BR resistance validated in the wild-type GL rather than in the susceptible variety LY?

The order of descriptions for the images in a Figure should correspond to their arrangement. For example: in Figure 1, image a and image b need to be swapped in position; in Figure 2, 2h is described first, so 2h should be labeled as 2a.

Author Response

Comment 1: Lines 134-136: Why the phenylpropanoid biosynthesis pathway was found to be only enriched in the five groups (R0 VS S0, R24 VS S24, R0 VS R24, S0 VS S24 and S0 VS S48), why not in another five groups? Please explain and discuss it.

Response: Thank you for your feedback. Different resistant plants have certain variability, and also according to the electron microscope observation of Ma et al. [21], the pathogens gathered on the plant leaves at 24 hpi, and the pathogens proliferated and invaded the plants at 48 hpi, so it is very critical time nodes at 0h, and 24, 48 hpi. We have revised the contents in line 144-148, 593-596.

Comment 2: Has the function of Bol4CL2 in BR resistance been verified, given that similar to Bol4CL41 this gene also shows significant expression differences between resistant and susceptible materials?

Response: Thank you very much for your question. When the gene was amplified in cabbage, Bol4CL41 was the first to be amplified. Due to the workload and the cost of genetic transformation of cabbage, only the genetic transformation seedlings of Bol4CL41 were constructed, without verifying the function of Bol4CL2 in the resistance to black rot. I hope we can get your understanding.

Comment 3: Why is the function of Bol4CL41 in BR resistance validated in the wild-type GL rather than in the susceptible variety LY?

Response: You asked a particularly good question. In cabbage, the GL is the easiest variety to transform genetically, while the sensitive variety LY and the resistant variety MY are very difficult to transform genetically. Therefore, we verified the function of genes in the wild-type GL.

Comment 4: The order of descriptions for the images in a Figure should correspond to their arrangement. For example: in Figure 1, image a and image b need to be swapped in position; in Figure 2, 2h is described first, so 2h should be labeled as 2a.

Response: Your suggestion is excellent. We have finished changing the order of the pictures.

Reviewer 2 Report

Comments and Suggestions for Authors
  • Figure legends often provide only surface-level descriptions without guiding readers toward key takeaways or interpretations.

  • Improving figure clarity and enhancing legend detail would help readers better understand the significance of the data.

  • Reference 1 and Reference 7 are duplicates. This should be corrected during revision.
Comments on the Quality of English Language
  • Several sentences are awkward or grammatically incorrect (e.g., “This makes them excellent research materials…” could be phrased more naturally).

  • There are noticeable errors in subject-verb agreement and tense usage.

  • Overuse or misuse of connectors and prepositions weakens the logical flow of certain paragraphs.

Author Response

Comment 1: Figure legends often provide only surface-level descriptions without guiding readers toward key takeaways or interpretations.Improving figure clarity and enhancing legend detail would help readers better understand the significance of the data.

Response: Your suggestion is excellent. We have modified the descriptions of the figures to summarise the key findings, for example, the annotation of Figure 1 has been changed to the phenotypes of MY and LY varied greatly after inoculation. The annotation of Figure 2 was changed to The phenylpropanoid biosynthesis pathway is differentially expressed in cabbage after inoculation. In the meantime, I’ve uploaded the high-resolution figures. Since inserting figures into documents reduces clarity, we have submitted them to the journal in pdf format.

Comment 2: Reference 1 and Reference 7 are duplicates. This should be corrected during revision.

Response: Thank you very much for your careful revision, we have corrected the references.

Comment 3: Several sentences are awkward or grammatically incorrect (e.g., “This makes them excellent research materials…” could be phrased more naturally).

There are noticeable errors in subject-verb agreement and tense usage.

Overuse or misuse of connectors and prepositions weakens the logical flow of certain paragraphs.

Response: Thank you for your feedback. We’ve made formatting and grammatical edits throughout the manuscript, with the changes highlighted in yellow.
